# Fossorial adaptations in African mole-rats (Bathyergidae) and the unique appendicular phenotype of naked mole-rats

Germán Montoya-Sanhueza [1,2 ✉], Gabriel Šaffa[1], Radim Šumbera[1], Anusuya Chinsamy[2], Jennifer U. M. Jarvis[2] & Nigel C. Bennett[3]

Life underground has constrained the evolution of subterranean mammals to maximize digging performance. However, the mechanisms modulating morphological change and development of fossorial adaptations in such taxa are still poorly known. We assessed the morpho-functional diversity and early postnatal development of fossorial adaptations (bone superstructures) in the appendicular system of the African mole-rats (Bathyergidae), a highly specialized subterranean rodent family. Although bathyergids can use claws or incisors for digging, all genera presented highly specialized bone superstructures associated with scratch-digging behavior. Surprisingly, *Heterocephalus glaber* differed substantially from other bathyergids, and from fossorial mammals by possessing a less specialized humerus, tibia and fibula. Our data suggest strong functional and developmental constraints driving the selection of limb specializations in most bathyergids, but more relaxed pressures acting on the limbs of *H. glaber*. A combination of historical, developmental and ecological factors in *Heterocephalus* are hypothesized to have played important roles in shaping its appendicular phenotype.

[1] Department of Zoology, Faculty of Science, University of South Bohemia, Branišovská 1760, České Budějovice 37005, Czech Republic. [2] Department of Biological Sciences, University of Cape Town, Private Bag X3, Rhodes Gift 7701, Cape Town, South Africa. [3] Department of Zoology and Entomology, Mammal Research Institute, University of Pretoria, Pretoria, South Africa. ✉email: g.montoya.sanhueza@gmail.com

One of the most intriguing questions in evolutionary biology pertains to how morphological diversity evolves. Subterranean mammals provide a classical model of convergent evolution because they share cranial and postcranial specializations that allow them to maximize the excavation of burrow systems[1,2]. Compared to their non-fossorial relatives, their limb bones are under strong selective pressures particularly modeled by digging kinematics, exhibiting increased bone robustness, enlarged areas for muscle attachment, and formation of novel characteristics such as fusion of bones and sesamoid hypertrophy[1–4]. Such adaptations respond to complex morpho-functional interactions between the individual and its surrounding medium to principally: (i) minimize the energetic costs of both loosening and transporting soils[5,6]; (ii) reduce fracture risks by increasing cortical thickening, and therefore bone resistance to muscular action and soil hardness[7–9]; (iii) increase body stabilization and anchorage during digging[1,2]; and (iv) have efficient locomotor performance by possessing short and symmetrical limbs allowing bidirectional locomotion in narrow spaces and a dense medium[9,10]. Some of the best known fossorial adaptations are associated with the forelimb, particularly in the humerus and manus of talpids (Talpidae; Soricomorpha), which have highly derived morphologies amongst mammals[11,12], the ossified tendon of golden moles (Chrysochloridae; Afrosoricida), which are the only tetrapods developing a third bone in the forearm[13], and the protuberant and distally located deltoid tuberosity of the humerus of fossorial mammals[1,14–16]. The uniqueness of a bone is strongly defined by a set of superstructures or eminences of varying shapes and sizes scattered along their external surface, such as tubercles and trochanters, which serve for tendon and ligament attachment, so their positioning along the bone is crucial for musculoskeletal and biomechanical functionality[17]. Although the development of bone superstructures has been more studied in surface-dwelling taxa such as laboratory rodents[17], the morphological diversity and morphogenesis of the appendicular system of fossorial taxa including the African mole-rats (Bathyergidae) are still poorly known. A better understanding of such aspects in non-model organisms will help elucidate how these complex adaptations evolve.

Bathyergids are highly specialized subterranean rodents that spend most of their lives underground and build extensive and complex burrow systems[18,19]. Among bathyergids, only one genus (*Bathyergus*) is a scratch-digger that predominantly uses its long fore-claws to build burrows in sandy soils, whereas all other genera are chisel-tooth diggers that use primarily their highly procumbent incisors for burrowing in a variety of soils, alternating from sandy to highly compacted[19,20]. This family comprises a wide spectrum of social organizations ranging from solitary to highly social (so called "eusocial")[21–23], as well as a wide range of body sizes, ranging from ~35 g in the naked mole-rat *Heterocephalus glaber* to up to 2 kg in the Cape dune mole-rat *Bathyergus suillus*. The phylogenetic relationships of the extant genera are well-established[24–26], and the fossil record shows that this group was apparently more diversified and widely distributed in the past than in the present[27,28].

Although the ecology, physiology, and behavior of bathyergids are well-documented[19,29], their appendicular adaptations have been surprisingly ignored. Most of the studies of their skeleton have focused on cranial features[30–32]. Similarly, their fossil record is largely represented by cranial and dental material[28], so that little is known about their appendicular skeleton. Additionally, although a few recent studies have increased our knowledge on the limb bone anatomy of some bathyergids[9,33,34], a comparative assessment of their limb adaptations including all bathyergid genera is lacking.

Given such a set of biological and ecological characteristics, African mole-rats represent a unique group of mammals for the investigation of multiple questions in evolutionary biology. In this study, we assessed the morphological diversity and early development of limb bone superstructures associated with fossoriality using an extensive collection of mole-rats comprising all six bathyergid genera. We focused on four bone superstructures that, under certain selective pressures and levels of development, represent clear advantage in terms of fossorial performance: the deltoid tuberosity (DT) in the humerus, the olecranon process (OP) in the ulna, the third trochanter (TT) in the femur, and the distal fusion of the tibia and fibula (DFTFi). Utilizing a form-function approach, we hypothesized that the forelimb of bathyergids will exhibit contrasting levels of development of bone superstructures as a reflection of their distinct chisel-tooth and scratch-digging behavior: (i) scratch-diggers would exhibit more developed bone specializations including an enlarged and distally located DT and enlarged OP, thus favoring powerful parasagittal motion of forearms for downward thrust of the forefeet to break up the soil, while (ii) chisel-tooth digging genera would show less developed DT and OP, since their limbs are not primarily involved in loosening soils. Assuming a generalized function of the hind limb skeleton for both body stabilization and soil transport among fossorial mammals[9,35], an enlarged TT and well-developed DFTFi are expected for both scratch- and chisel-tooth diggers, i.e., less morphological variation will be present in the hind limb superstructures among bathyergid genera. Such morphology is then compared with the closest extant relatives of Bathyergidae, the Petromuridae, Thryonomyidae and Hystricidae. To determine when bone superstructures appear during ontogeny, we also examined the perinatal limb development of multiple bathyergid species. Additionally, we assessed how morphofunctional proxies associated with bone superstructures scale with body size, and based on our anatomical findings, we were also able to reconstruct the evolution of such characters using the well-known phylogenetic framework existing for this family.

## Results and discussion
The main ecological and morphological characteristics of African mole-rats studied here are presented in Table 1.

### Comparative anatomy
*Humerus*. Both scratch-diggers and chisel-tooth diggers (except *H. glaber*) exhibit a similar humeral phenotype with a well-developed (projected) and distally located DT in the anterolateral side of the diaphysis (Fig. 1a and Supplementary Table 1). The DT in newborns is cartilaginous and has a relatively similar shape and position as compared with adults. No cartilaginous tissue is observed in the DT of juveniles (Fig. 2a–f). Thus, the DT grows initially by endochondral ossification, and increases in size by periosteal bone formation later during juvenile stages. In *H. glaber*, the DT is reduced to a poorly developed (non-projected) and sometimes indistinguishable structure in the midshaft, although a small scar for the insertion of the *mm. deltoidei* appears in adults (Fig. 1a). No cartilaginous DT was observed in newborns of *H. glaber* (Fig. 2g). Some specimens of *H. glaber* showed a poorly developed deltopectoral crest (DC) running all the way down through the anterolateral side of the shaft and joining the DT scar area around the midshaft (Fig. 1a).

The major similarities among adult bathyergids are: (i) having a large and ellipsoidal humeral head (HH), which is highly convex in lateral view, although the HH of *H. glaber* has a small and flattened articular morphology (Supplementary Fig. 1); (ii) the trochlea and capitulum have a well-defined form and size, although these are poorly defined (rudimentary) in most *H.*

**Table 1 Digging mode, social system, family size (mean/max.), body mass (mean), and sample size (*n*) of African mole-rats (Bathyergidae) analyzed in this study.**

| Species | *n* | Digging mode | Social system | Family size | Body mass (g) |
|---|---|---|---|---|---|
| *Bathyergus suillus* | 78 | Scratch-digging | Solitary | – | 866.85[a] |
| *Bathyergus janetta* | 6 | Scratch-digging | Solitary | – | 384.5[a] |
| *Heliophobius argenteocinereus* | 38 | Chisel-tooth | Solitary | – | 176[a] |
| *Georychus capensis* | 51 | Chisel-tooth | Solitary | – | 180.5[a] |
| *Cryptomys hottentotus* | 53 | Chisel-tooth | Social | 5/14 | 56.3[a] |
| *Fukomys mechowii* | 32 | Chisel-tooth | Highly social | 11/20 | 480.8[a] |
| *Fukomys damarensis* | 48 | Chisel-tooth | Highly social | 12/41 | 140[a] |
| *Heterocephalus glaber* | 76 | Chisel-tooth | Highly social | 75/300 | 33.9 |

Ecological data obtained from multiple sources (see Supplementary References).
[a]Body mass averaged for males and females.

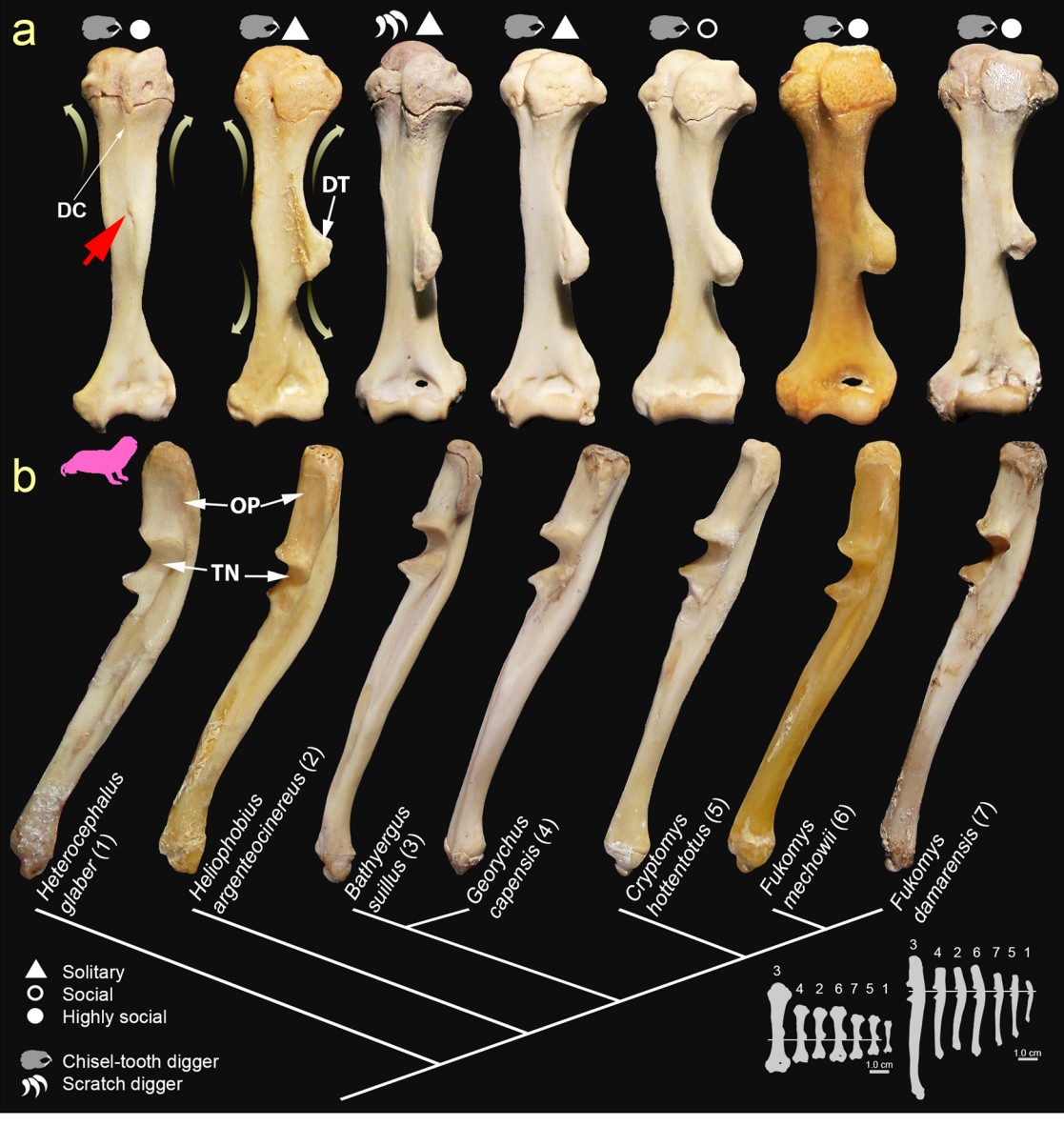

**Fig. 1 Forelimb phenotype (humerus and ulna) and phylogeny of African mole-rats including main digging mode and social system. a** Anterior view of humeri showing the projected deltoid tuberosity (DT) in all genera, except in *Heterocephalus glaber* (pink silhouette) that has a poor development of this feature, and a small scar for the insertion of the *mm. deltoidei* (red arrow head). Some specimens also showed a poorly developed deltopectoral crest (DC). Yellow arrows indicate the areas of bone thickening observed during ontogeny in all species, except for the distal portion of *H. glaber*, which remains narrow at the distal end. **b** Lateral view of ulnae showing the enlarged olecranon process (OP) and cortical thickening of the diaphysis under the trochlear notch (TN) in all species. Bone silhouettes show the real size of each bone among species, and correspond to the largest specimens of each species analyzed here. Phylogenetic relationships adapted from Upham et al.[79]

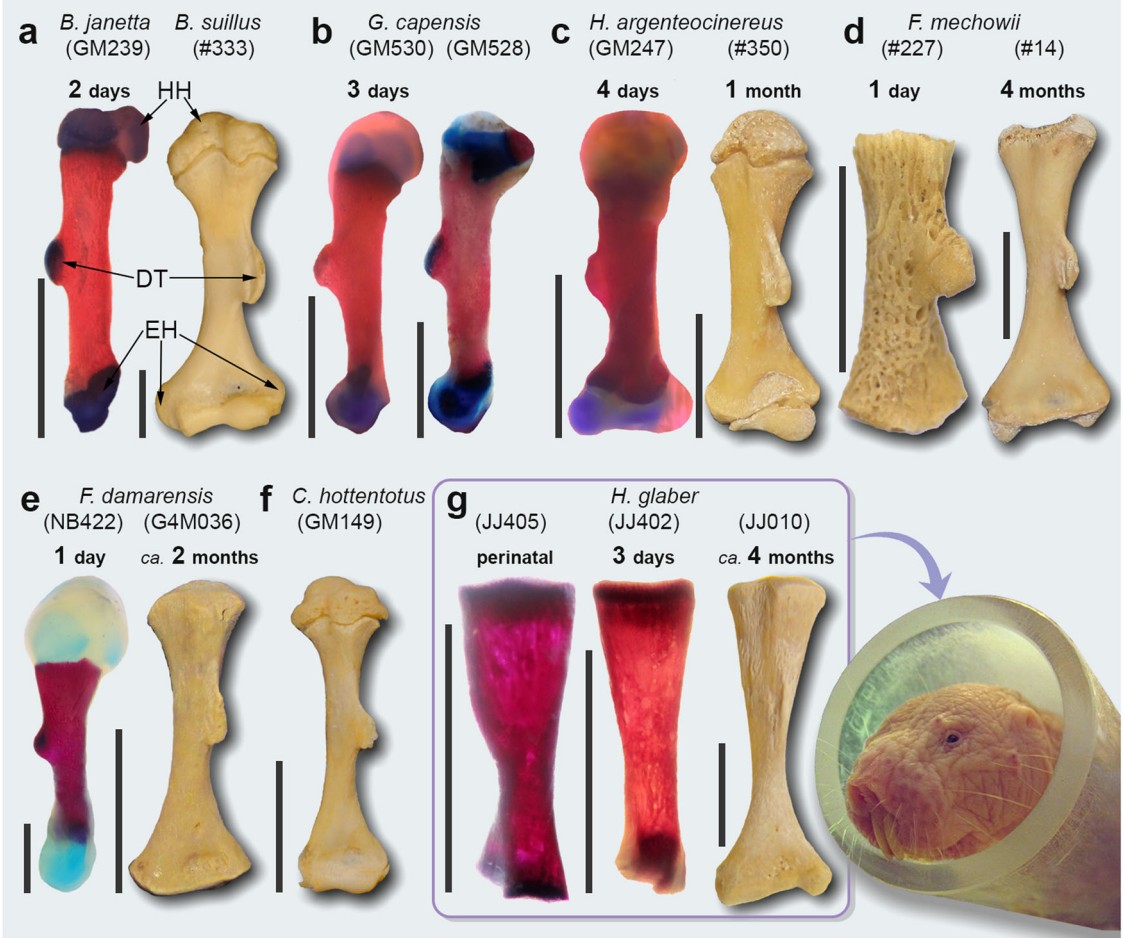

**Fig. 2 Early postnatal development of the humerus of African mole-rats. a** Well-developed deltoid tuberosity (DT) in *Bathyergus suillus* and *B. janetta* originated from a localized primordium in the diaphysis. Chondroepiphyses show an enlarged humeral head (HH) and wide cartilaginous epicondyles (EH). Similar development of humerus in **b** *Georychus capensis*, **c** *Heliophobius argenteocinereus*, **d** *Fukomys mechowii*, **e** *F. damarensis*, and **f** *Cryptomys hottentotus*. **g** Note the absence of cartilaginous tissue in the diaphysis of *Heterocephalus glaber*. Scale bars represent 5.0 mm, except in specimens JJ402, JJ405, JJ010, and #227, which represent 2.5 mm. Stained specimens are in lateral view, except GM247 which is in posterolateral view. Non-stained specimens are shown in anterior view.

*glaber* specimens. Some important morphological changes observed during ontogeny are: (i) pups and juveniles have relatively more robust chondroepiphyses as compared to the epiphyses of adults, so that the HH and epicondyles are conspicuously wider than the diaphysis; (ii) mediolateral diaphyseal thickening occurs mainly towards both epiphyses, so that the diaphysis is relatively symmetrical in proximal and distal aspects, except in the distal diaphysis of *H. glaber*, which is relatively narrow.

The closest extant relatives of bathyergids, *Hystrix africaeaustralis* (Hystricidae) and *Petromus typicus* (Petromuridae) exhibited a conspicuous DT, with *H. africaeaustralis* having well-developed DT and DC in the midshaft of the diaphysis which extend proximally, while *P. typicus* has a less developed and more proximally located DT (Supplementary Table 1, see below).

*Ulna.* All bathyergids show a long and mediolaterally narrow ulnar phenotype, with no conspicuous shape changes appreciated during ontogeny. An enlarged OP is observed in both scratch-diggers and chisel-tooth diggers (Fig. 1b and Supplementary Table 1), and is already observed in newborns, although still cartilaginous. Pups also exhibit cartilaginous anconeal and coronoid processes, as well as a trochlear notch, which become completely ossified in juveniles. Apart from the conspicuous

elongation of the ulna, the diaphysis of all species is thicker predominantly in its anteroposterior axis, particularly below the trochlear notch.

Both *H. africaeaustralis* and *P. typicus* have a conspicuous OP, although this is less developed in the latter species (Supplementary Table 1, see below).

*Femur.* All specimens from either scratch-digging or chisel-tooth digging genera show a similar femoral phenotype, with a well-developed TT that extends from the proximal section of the diaphysis until just above the midshaft (Fig. 3a and Supplementary Table 1). The TT is already observed in newborns as a small protuberance composed of cartilaginous tissue, which ossifies gradually during juvenile stages. In adults, the TT exhibits considerable intraspecific variation, sometimes localized to a particular area and others having a longer extension along the diaphysis. Proximal and distal chondroepiphyses of pups and juveniles of all bathyergids are considerably enlarged with respect to their thinner diaphysis. Importantly, *H. glaber* exhibits high variation in the morphology of the proximal epiphysis, which is more robust in some specimens than in others, particularly by having a shorter femoral neck, and therefore a more constricted morphology and fused femoral head and greater trochanter (Fig. 3a; see below).

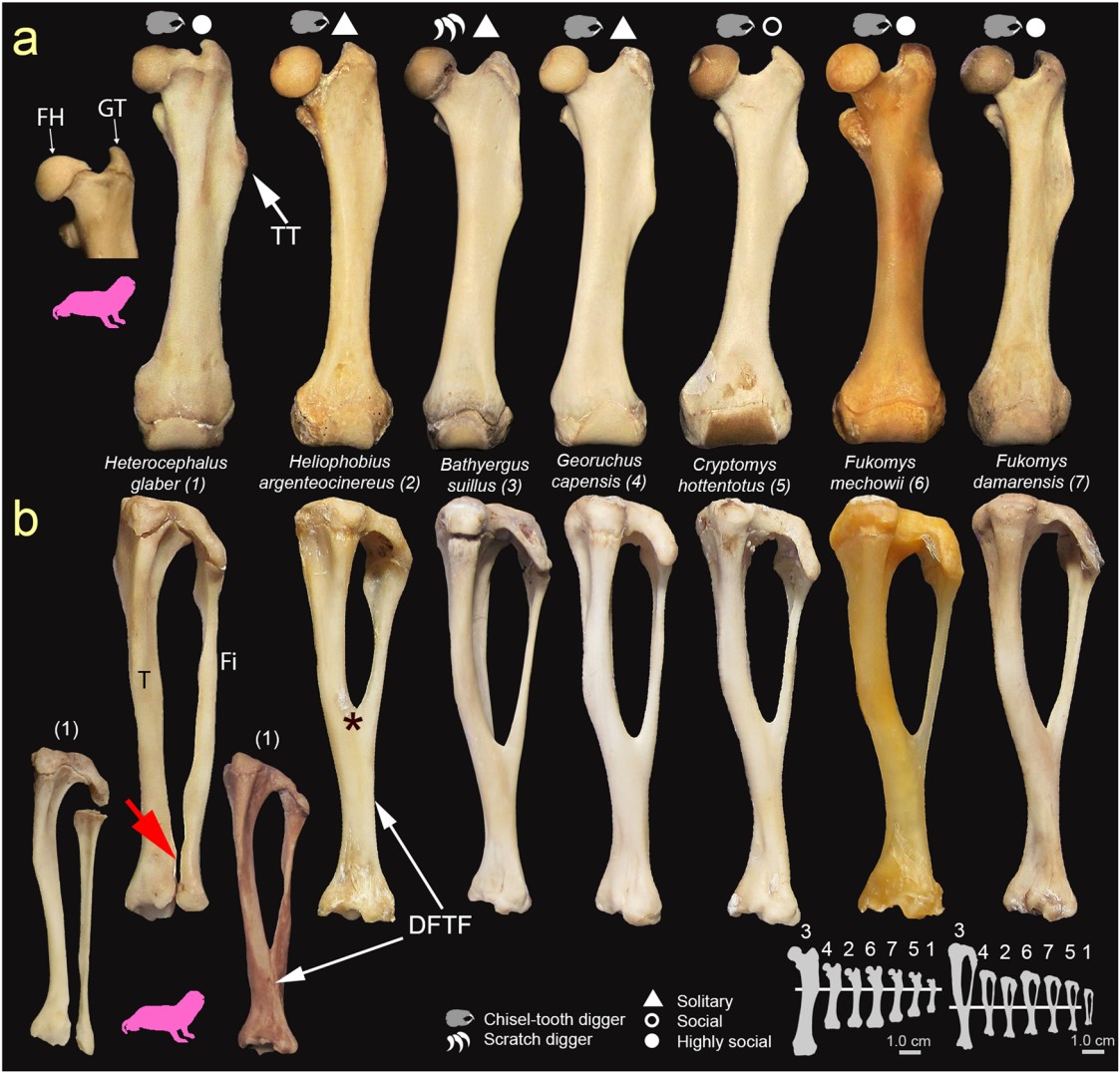

**Fig. 3 Hind limb phenotype (femur and tibia-fibula) of African mole-rats. a** Anterior view of the femur showing the well-developed third trochanter (TT) in both scratch-digging and chisel-tooth digging genera. Note the variable condition (separated and fused) of the femoral head (FH) and greater trochanter (GT) in *Heterocephalus glaber* (pink silhouette). **b** Anterior view of the tibia (T) and fibula (Fi) showing distal fusion (DFTFi) and junction point (*) in most species, as well as the non-ossified condition of *H. glaber* (red arrow). Two additional specimens of *H. glaber* (pink silhouette) are included to show the complete proximal and distal separation of T and Fi (left) and the exceptional fusion of such bones (right) (bones not to scale). Bone silhouettes show the real size of each bone among species, and correspond to the largest specimens of each species analyzed here.

Both *H. africaeaustralis* and *P. typicus* have a highly reduced TT (Supplementary Table 1, see below).

*Tibia-fibula.* In all species, the tibia is considerably more robust than the fibula (Fig. 3b), although such differences are much less marked at birth. At birth, the tibia is straight and undifferentiated, becoming slightly curved internally around the midshaft during ontogeny. In newborns, the fibula of all species is rather straight and thin, and located posteromedially with respect to the tibia. The adults of all species show a similar development of the tibia and fibula, although such process differed notably in *H. glaber*. A few days after birth, these bones form a complex structure where their proximal and distal epiphyses ossify and fuse, thus forming the DFTFi (Fig. 3b and Supplementary Table 1). In general, the DFTFi extends from the lower section of the midshaft and progresses until the distal epiphysis during early ontogeny. In adults, the proximal extension of this fusion, which results in the formation of the tibio-fibular junction, locates

around the midshaft. The beginning of the distal fusion seems to be variable among species: some species show perinatal fusion (one day old in *F. damarensis*, and two days old in *B. janetta*), whereas other species show it later (three days old in *F. mechowii*). Likewise, intraspecific variation for this trait was also observed: some newborns of *F. damarensis* exhibited a fused condition (e.g., #NB422, #G3M036), while others still had unfused bones (e.g., #NB423, #G3F035). *Heterocephalus glaber* showed a distinct condition where only the proximal epiphysis ossifies and fuses, while the distal region of these bones remains unfused and unites only by a syndesmotic joint of connective tissue at the tip of the distal epiphysis (Fig. 3b). Exceptionally in our sample, three adults of *H. glaber* (out of 70 adults) showed a true distal ossification of the tibia and fibula (Fig. 3b and Supplementary Table 1).

The distal tibia and fibula of *H. africaeaustralis* and *P. typicus* are unfused, thus lacking a DFTFi (Supplementary Table 1, see below).

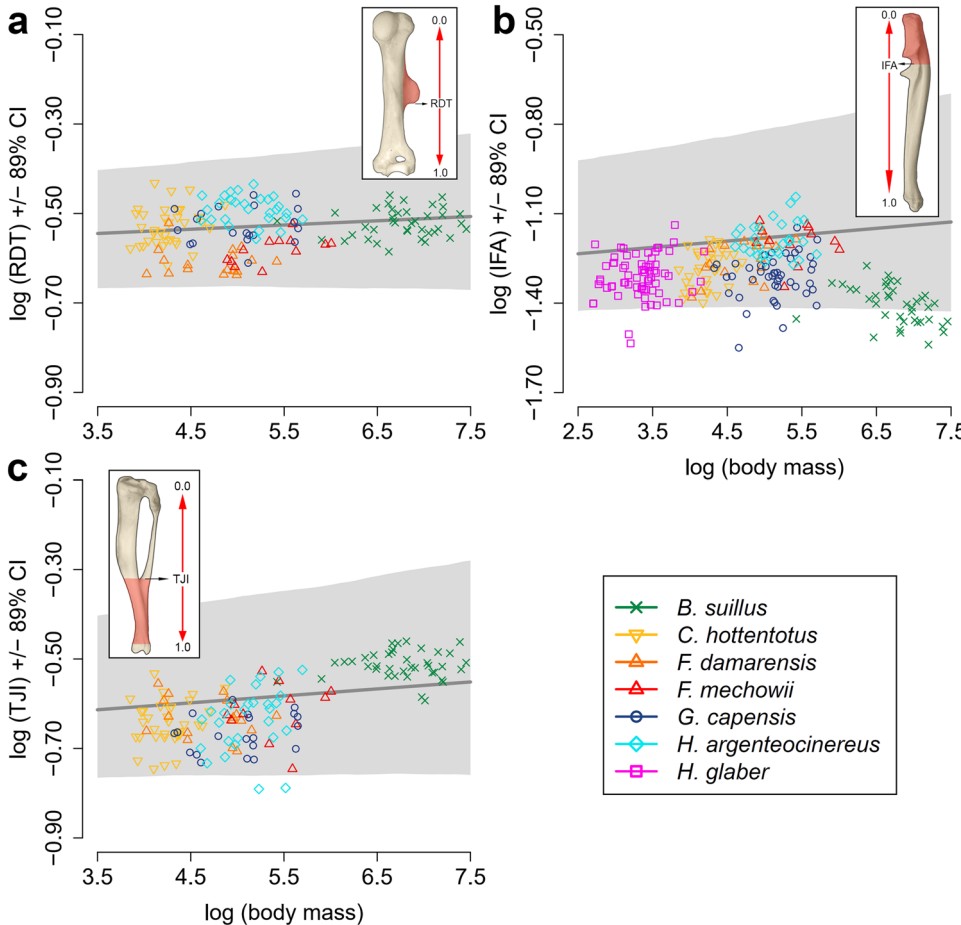

**Fig. 4 Scaling effects of three morpho-functional indices with body mass. a** Relative position of the deltoid tuberosity (RDT); **b** Index of fossorial ability (IFA); and **c** Tibio-fibular junction index (TJI). Solid lines indicate the scaling effect with BM, whereas shaded areas are 89% compatibility intervals (CI).

**Effects of body mass and phylogeny on morpho-functional indices.** Given the wide range of body sizes among bathyergids, we assessed how functional proxies associated with bone super-structures scale with body mass (BM). Three morpho-functional indices were analyzed: the relative position of the DT (RDT); the index of fossorial ability (IFA); and the tibio-fibular junction index (TJI); all of which represent the degree of development of the DT, OP, and DFTFi, respectively. Model comparison revealed that for RDT and TJI, a model without BM makes slightly better prediction than model with BM, as indicated by WAIC (Supplementary Table 2). Essentially, both types of models –with and without BM– would make almost identical predictions. In contrast, for IFA, model with BM shows better predictive performance than model without BM ($\Delta_i$ WAIC = 6.31; Supplementary Table 2). However, approximate standard error of the difference between the two models, $\Delta_i$ SE, is 4.43 (Supplementary Table 2), suggesting considerable uncertainty in the degree of improvement in prediction when including BM into the model. Similarly, both regressions with population-level estimates (Fig. 4) or individual estimates of intercept and slope for each species, showed essentially no, or only weak, scaling effect with BM for all morpho-functional indices (Supplementary Tables 3 and 4). Lastly, there is very low contribution of phylogeny to the between-species variability among both intercepts and slopes, with more variability attributed to species-specific effects in all three indices (Supplementary Table 3). This is not surprising given the low number of species in our sample, all belonging to a single family.

**Ancestral state reconstructions.** Stochastic character mapping showed that the projected DT, present in all taxa (except *H. glaber*) was with 83% posterior probability (PP) present in the common ancestor of Petromuridae and Thryonomyidae, with only 63% PP present in the common ancestor of Bathyergidae, while it was lost (or reduced) in a lineage leading to *Heterocephalus* (Fig. 5a). However, since *Heterocephalus* is a basal lineage among Bathyergidae, it likely introduces uncertainty to the estimates of ancestral states at deeper nodes. For example, the common ancestor of Bathyergidae shows 65% PP of having DT, while the PP at the root is only 47% (Fig. 5a). The posterior probability of the enlarged TT being present in the common ancestor of African mole-rats is 69%, and considering its absence in Petromuridae, Thryonomyidae, and Hystricidae, it may represent a synapormorphy of the family Bathyergidae (Fig. 5b). The DFTFi was most likely present in the common ancestor of *H. argenteocinereus* and the rest of the bathyergids (98% PP), with only *H. glaber* showing a shared trait with thryonomyids, petromurids, and hystricids, i.e., lack of DFTFi (Fig. 5c).

**Fossorial adaptations in Bathyergidae.** The comparative analysis of multiple limb bones, as well as the assessment of bone superstructures and morpho-functional indices allowed us to uncover the morphological diversity of African mole-rats. In general, regardless of the digging mode, social system and body size, most African mole-rats have a well-developed set of appendicular fossorial adaptations, such as a projected and

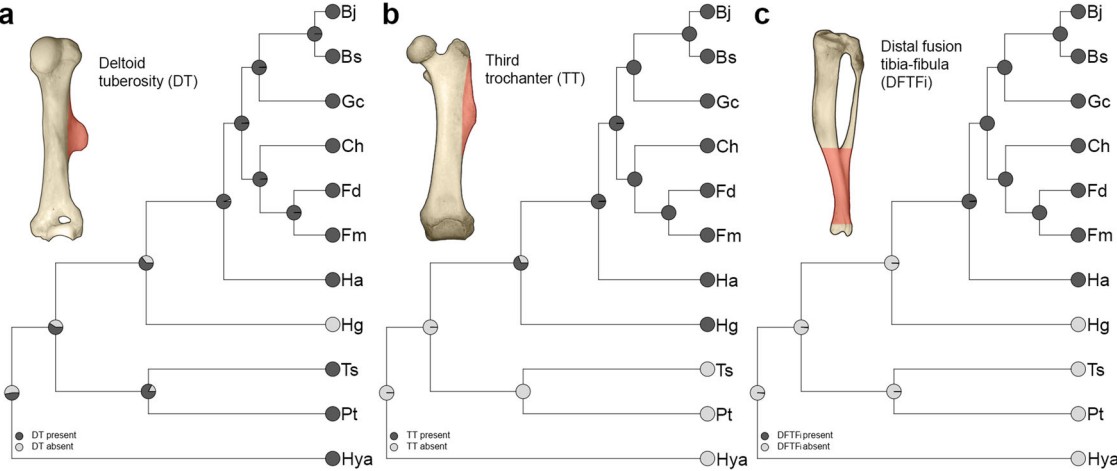

**Fig. 5 Ancestral state reconstructions of discrete bone superstructures in Bathyergidae and the outgroups Petromuridae, Thryonomyidae, and Hystricidae. a** Deltoid tuberosity (DT); **b** third trochanter (TT); and **c** distal fusion of the tibia-fibula (DFTFi). Values at internal nodes were obtained by simulating character maps over the posterior sample of 1000 trees, using a method of stochastic character mapping. Simulated values are interpreted in terms of posterior probability. Bone illustrations show the analysed bone superstructure (red area). Bs *Bathyergus suillus*, Bj *Bathyergus janetta*, Ch *Cryptomys hottentotus*, Fd *Fukomys damarensis*, Fm *Fukomys mechowii*, Gc *Georychus capensis*, Ha *Heliophobius argenteocinereus*, Hg *Heterocephalus glaber*, Hya *Hystrix africaeaustralis*, Pt *Petromus typicus*, Ts *Thryonomys swinderianus*. Phylogenetic relationships obtained from www.vertlife.org[79].

distally located DT, enlarged OP, extended TT, and an ossified DFTFi (Figs. 1, 3, and 4). This agrees with previous reports in bathyergids[9,33,34]. However, this first ever comparative approach to assessing the limb phenotype of all African mole-rat genera, allowed us to determine the unique humeral and tibio-fibular morphology of *H. glaber*, which differed considerably from other bathyergids by lacking a projected DT (Fig. 1a) and DFTFi (Fig. 3b). *Heterocephalus glaber* also possess variable morphology of the proximal femur, which differs from the main condition of other bathyergids (Fig. 3a; see full description in Montoya-Sanhueza et al.[36]).

The presence of well-developed bone superstructures (i.e. DT, OP, TT, and DFTFi) in *Bathyergus* is concordant with similar findings in many other fossorial mammals. Such features represent clear adaptive advantages to produce large out- and in-forces with the forelimbs, resist the tendency to move backward/rearwards with the hind limbs, and increase bone resistance to bending and torsional loads during burrowing (refs. [1,3,7–9,14–16]; Supplementary Table 5), thus confirming our hypothesis for this scratch-digging genus. However, the presence of similar adaptations in the limbs of both solitary (*Georychus* and *Heliophobius*) and social (*Cryptomys*, *Fukomys* and *Heterocephalus*) chisel-tooth diggers does not support our functional predictions of reduced limb specializations for these taxa, which it may suggest an equally important role of their limbs for burrowing as compared to scratch-diggers. This suggests that certain limb and neck muscles may also have an important role during chisel-tooth digging for body anchorage and force production. The analysis of digging sequences in *Fukomys micklemi*, a chisel-tooth digger, and *Myospalax myospalax* (Spalacidae), a scratch-digger/head lifter provides insightful information about their digging behavior[37,38]. The neck and shoulder musculature of these species are fundamental for limb stabilization and anchorage to efficiently produce forward force with the incisors, jaws and head to go up through the soil by using downward forelimb thrust, i.e. extension of the elbow[37,38]. Similar function of hind limbs aiding with forward movement of the body can be expected during chisel-tooth digging[2,35,37]. In fact, a combination of opposing forces are generated during digging, where the head and neck push upward against the roof of the tunnel, and hind limbs push downward, as well as the forelimbs

push forward against the medium while the hind limbs push backward against the floor of the tunnel[35]. Thus, the muscles responsible for downward forelimb thrust during parasagittal scratch-digging are not uniquely useful for loosening the soil, but also for pushing up and throwing off the substrate during chisel-tooth digging and head lifting digging[1,37,38]. This indicates that musculoskeletal specializations favoring parasagittal scratch-digging may also have a selective advantage in chisel-tooth diggers (and vice versa). It is possible that the appendicular phenotype of African mole-rats, and probably also other fossorial taxa (irrespective of their primary digging mode), is under strong selective pressures related to the complexity of burrowing, where one limb can perform multiple tasks during the entire digging process. Further analysis of the digging kinematics and musculature of African mole-rats is fundamental to determine the specific relationship between certain digging behaviors and morphology. Nevertheless, the presence of discrete bone super-structures (i.e., projected DT, enlarged OP, extensive TT, and DFTFi) in mole-rats may not represent exclusive functional proxies of their primary digging mode, an aspect that may be relevant for the reconstruction of specific digging modes of extinct taxa.

Our results also demonstrate that the development of bone superstructures in African mole-rats is the result of different morphogenetic mechanisms. The invariable presence of a cartilaginous DT, OP, and TT in newborns and pups of all species (except *H. glaber* perinatals lacking a DT primordium) (Fig. 2) indicates an endochondral and perinatal origin for such features, like the development of bone superstructures of other rodents[17]. The exact timing of development of bone super-structures in mammals are not completely known, although the expression of an ossified DT in mice occurs prenatally during early skeletogenesis, whereas the ossification of the OP and TT occurs later during perinatal age[17,39,40]. Although muscle activity in combination with the genetic program are known to regulate the attainment of optimal bone shapes during embryogenesis[41,42], endochondral bone formation is usually linked to strong genetic regulation and usually exhibits relatively stable development. Thus, DT, OP, and TT in bathyergids are unlikely to have originated by biomechanical stimulation. On the contrary, the process of fusion of the tibia and fibula, occurs by

intramembranous ossification several days after birth (e.g. in mice [17,43,44]). This process, although yet still not fully understood, it is hypothesized to result from more genetically independent processes and postnatal biomechanical stimulation (e.g., intermittent pressure and tension) exerted on periosteal surfaces that stimulate subsequent bone (re)modeling[43]. Thus, external factors may be morphogenetically primary for the shaping of the DFTFi in mole-rats. In our study, the development of the DFTFi also seems to be more temporally variable among species than other features, with bone fusion generally occurring several days after birth (Supplementary Table 1). The fusion begins at the distal thirds of the tibia and fibula (e.g., *F. damarensis*). In *H. glaber*, such bones are united by a syndesmotic joint at the same region of fusion of other species, although these usually do not fuse. This suggests a close relationship between the formation of the ligament in this region of the diaphysis and the initiation of the fusion process. The exceptional finding of three specimens of *H. glaber* with a distal fusion of the tibia and fibula around the ligament region (Fig. 3b and Supplementary Table 1) provides further support to this idea. A similar mechanism has been proposed for the fusion of these bones in tenrecids, which exhibit variable modes of locomotion and whose bones fuse over the area generally occupied by the inferior interosseous ligament[45]. These data support a biomechanical hypothesis for the origin of the DFTFi in bathyergids, and highlights the role of functional demands, external loading and probably muscle activity as potential regulators for the initiation of tibio-fibular fusion in these animals. Additional observations of ontogenetic sequences are required to fully understand this process.

The developmental disparity observed in bathyergids demonstrates that the morphogenesis of fossorial adaptations encompasses the coupling of different mechanisms acting during prenatal formation (early skeletogenesis) and postnatal bone (re)modeling. Similar developmental disparity of limb elements has been suggested for Cape dune mole-rats (Bathyergidae)[9], and other fossorial taxa, such as water voles (Arvicolinae)[46] and tuco-tucos (Ctenomyidae)[47], as well as for non-fossorial mammals such as laboratory rodents and bats[43,44]. These features among the Bathyergidae are of crucial interest for the understanding of phenotypic evolution in subterranean and other mammals.

**The limb phenotype of naked mole-rats**. The most intriguing finding of our study is the absence of two typical fossorial adaptations in the limbs of *H. glaber*, the projected DT and DFTFi (Figs. 1a, 3b, and 6). Such phenotype was also reported for wild individuals of *H. glaber*[48,49], although never assessed under a comparative approach including all bathyergid genera. A closer examination of the humeral anatomy of all other fossorial rodent taxa demonstrates that the projection of the DT is present in all species, regardless of their digging mode and social system (Fig. 6a and Supplementary Table 6), although not exclusive to subterranean taxa. This makes *H. glaber* standing out among bathyergids and other fossorial rodents, and raises the question of why this highly subterranean species did not develop such limb specializations, and actually rather reduced its humeral specialization for digging.

A considerable variation in DT and DC morphology has been observed in other fossorial mammals, the Orycteropodidae (Tubulidentata). The humerus of the extant fossorial aardvark *Orycteropus afer* shows a projected DT and DC, while *Orycteropus abundulafus*, from the Mio-Pliocene of northern Chad has a slender humerus lacking them[50]. In functional terms, a highly reduced DT suggests a diminished power stroke for the flexion of the arm, thus reducing the strength of parasagittal digging (refs. [1–3,14–16]; Supplementary Table 5). The humerus of

*H. glaber* also differs from other extant bathyergids by having a rather small and flat humeral head, and an extremely narrow distal diaphysis. A recent study including a few sexually mature naked mole-rats also describe a poorly defined proximal forearm anatomy with reduced areas for muscle attachment[34]. A comparatively thinner distal diaphysis and less defined distal epiphysis suggest lower bending resistances of the humerus and relatively smaller muscles involved in scratch-digging behavior, respectively. A flattened humeral head is likely compromising the stabilization and shock absorption area of the shoulder needed for digging[15], although it probably allows a higher degree of arm rotation[51]. In this sense, the humeral phenotype of *H. glaber* may facilitate a wide range of motion in the humero-scapular joint, thus resembling that of unspecialized mammals with predominantly surface-dwelling or ambulatory locomotor modes, such as some tenrecids[51]. Similarly, the lack of a DFTFi in *H. glaber* suggests a considerable reduction in tibial and hind feet robustness (Supplementary Table 5), although it may also confer an increased range of zeugopodial and autopodial motion, as observed in surface-dwelling mammals with ambulatorial and arboreal locomotion[43,52,53]. The ultimate causes of such reduced level of fossoriality in the limbs of *H. glaber* are intriguing, and a series of factors might be involved in its evolution.

*Heterocephalus glaber* primarily uses its incisors to excavate soils, which are usually compacted and dry for most of the year[20,54–56]. This is important because the mechanical advantage and forces exerted by cranial specializations such as tooth-digging are considerably greater as compared to forelimb-claw specializations[3]: incisors are covered by hard enamel and rooted into the skull and mandibles, while claws are composed of a softer material (keratin) and attached to flexible digits[57]. Although *H. glaber* and other chisel-tooth digging bathyergids have a specialized masticatory musculature[58], *H. glaber* produces the strongest relative biting forces within the family[59]. This set of dental and cranial adaptations may have prevented the development of further limb specializations in *H. glaber*. Indeed, *H. glaber* has the most reduced claws in manus and pes among bathyergids (Fig. 6b), suggesting a considerable relegated function of limbs to break up soils[60]. However, *H. glaber* also possesses appendicular adaptations aiding with digging, including an enlarged OP, extended TT and long bones with thick cortical walls[61], which agrees with behavioral observations in wild and captive individuals reporting the vigorous use of their fore- and hind limbs for removing and transporting soils[20,54,60,62]. Considering the latter and the fact that limb musculature is strongly involved during chisel-tooth digging (see previous section), it is unlikely that the primary digging mode of *H. glaber* may be the only factor precluding further specialization of their limb bones.

**Evolution of fossorial adaptations in Bathyergidae and naked mole-rats**. The ancestral reconstructions including the non-subterranean closest living relatives of bathyergids, the Petromuridae, Thryonomyidae, and Hystricidae, may help understand the evolution of fossorial adaptations in this family and the specific phenotype of *H. glaber* (Fig. 5). *Heterocephalus* is the most basal lineage of the Bathyergidae[24–26], with recently updated diverging times dated to the Oligocene (29.02 Ma)[26]. A subsequent divergence in the middle Miocene (13.37 Ma) separated *Heliophobius* from the rest of the bathyergids[26].

Our data shows that the appendicular evolution of bathyergids is marked by clear morphological changes in the hind limb (Fig. 5), with the extended TT and DFTFi representing synapomorphic features in this family, since *P. typicus*, *H. africaeaustralis* and *H. indica* show highly reduced TT and unfused tibia and fibula. Systematic studies already suggested that

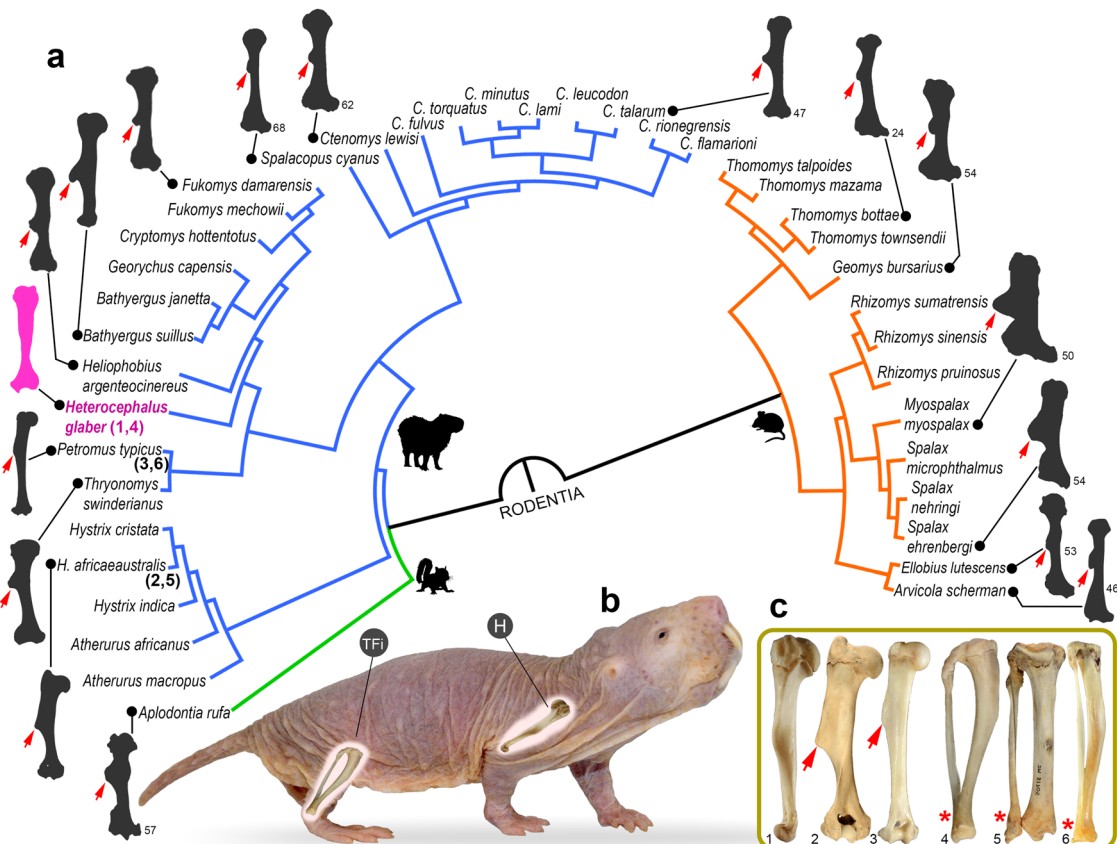

**Fig. 6 The limb phenotype of *Heterocephalus glaber* compared with other fossorial mammals. a** Phylogenetic relationships of 33 fossorial species from the three main rodent lineages illustrating the distinctiveness of the humerus of *H. glaber* (pink bone): Sciuromorpha (green branches), Hystricomorpha (blue branches), and Myomorpha (orange branches). Note the presence of a projected deltoid tuberosity (DT, red arrows) in all subterranean taxa, except in *H. glaber* where it is highly reduced (see details in Supplementary Table 6). Bones silhouettes not to scale. The bone silhouettes extracted and modified from other sources are indicated by a reference number (see details in Supplementary References). **b** Anatomical disposition of the humerus (H) and tibia-fibula (T-Fi) in *H. glaber*. Note the poor development of claws in the pes and manus in this species. **c** Comparison of the humerus (1,2,3), tibia and fibula (4,5,6) of *H. glaber* (1,4) and the non-subterranean outgroups, *Hystrix africaeaustralis* (Hystricidae) (2,5) and *Petromus typicus* (Petromuridae) (3,6). Note the presence of a projected DT in the closer relatives of Bathyergidae (2,3), the poor development of this feature in *H. glaber*, and the unfused condition (*) of the distal tibia and fibula in all taxa (4-6). Bones not to scale. Phylogenetic relationships obtained from www.vertlife.org[79].

bathyergids represented an exception among hystricognath rodents by possessing a DFTFi[63,64], although such studies overlooked the unfused condition of tibia and fibula in *H. glaber*. The latter provides further evidence for the basal placement of *H. glaber* within the Bathyergidae and suggests that the non-fusion of the tibia and fibula is rather plesiomorphic for the family, with the synapomorphic condition only attained more recently in derived genera (Fig. 5c). Yet, the presence of a few *H. glaber* specimens having a DFTFi complicates the scenario (see below). A different story can be told for the evolution of the forelimb. Among the phiomorphs analyzed, *P. typicus* (Fig. 6c) and *Thryonomys swinderianus*[65], as well as *H. africaeaustralis* and *H. indica* (Fig. 6c) also possess a relatively well-developed DT and OP, whereby such features represent plesiomorphic traits for Bathyergidae, while the highly reduced DT in *H. glaber* appears to be an autapomorphic event (Fig. 5a).

The extinct *Bathyergoides neotertiarius* (Bathyergoididae) from the early Miocene of East Africa and Namibia[27,28,66], one of the proposed ancestors of bathyergids may help us understand the appendicular evolution of early mole-rats. This is a medium-size rodent with a robust humerus and the ulna exhibiting an enlarged OP[27,66], like other phiomorphs analyzed here. Importantly, *B. neotertiarius* has a considerably reduced DT (although more

developed than in *H. glaber*), a less robust femur (with a reduced TT as compared to modern genera), and lack of DFTFi[27,66]. The general limb anatomy of this fossil clearly resembles the phenotype of non-fossorial phiomorphs (and *H. glaber*) rather than that of other more derived bathyergids, thus suggesting that the limb phenotype of *H. glaber* is more closely related to the ancestral condition. Based on such features, *B. neotertiarius* has been suggested to exhibit certain degree of burrowing behavior, but to a lower level as compared to extant bathyergids[27,66]. It is reasonable to hypothesize that early bathyergids had a comparatively more restricted use of subterranean niches, and that the evolution of the appendicular fossorial phenotype in Bathyergoidea (Bathyergoididae + Bathyergidae) occurred gradually, at least during their early evolution.

The existence of a fully fossorial limb phenotype in the solitary *H. argenteocinereus* marks an important event in the appendicular evolution of the family. A considerable increment in burrowing demands in this lineage, probably preconditioned by a well-developed forelimb anatomy (as seen in the outgroups and fossil forms), would have allowed the more stable occupation of subterranean habitats and therefore triggered the selection of novel skeletal phenotypes, such as highly projected DT and DFTFi. The transition to a solitary subterranean lifestyle in

*H. argenteocinereus* (assuming a more social condition for its ancestors) may have triggered stronger selective pressures for the selection of specialized bone superstructures, such as an enlarged TT and DFTFi. Additional fossil material will clarify this issue.

Our data of reduced humeral specializations and simplified tibia and fibula in *H. glaber* suggests the influence of developmental processes on their limb phenotype. The lack of a cartilaginous primordium in the humerus of perinatal individuals (Fig. 2) is probably associated with a distinct developmental variant of the DT not expressed during skeletogenesis. Likewise, the variable condition of the tibia and fibula in *H. glaber* may also represent a (facultative) developmental variant, which in part reinforces the idea that this species exhibits high developmental plasticity (ref. [36] and references therein). In developmental terms, phenotypic simplification has been attributed to reduced growth rates (neoteny) in paedomorphic species[67]. In this sense, it is known that pups of *H. glaber* possess the slowest somatic growth rates among bathyergids[62,68], which is further supported by recent assessments of their postnatal long bone growth patterns[60]. These data and a large body of information supports the neotenic condition of *H. glaber*[69,70], indicating that heterochronic processes may have played an important role in the morphological differentiation (and simplification) of this species.

Some factors that may have reduced the selection of a more specialized fossorial limb phenotype in *H. glaber* are (i) the formation of organized sequences of "cooperative" digging, where colony members work together in relay forming chains[20,54], (ii) the peculiar way of disposing soil known as "volcanoing"[20], and (iii) the larger number of individuals comprising the colony (Table 1). Because cooperative digging implies increased individual dynamism and flexibility to move over other individuals and push soil backwards over obstructed tunnels, it is likely that such behaviors may have reduced the need for the selection of overspecialized structures, and rather prioritize limb flexibility and dexterity. Nevertheless, the formation of digging sequences and kicking behavior may not be exclusive to *H. glaber*[29], so additional studies on the digging behavior of this and other bathyergids are critical to understand the energetic and functional implications of such activities. The much larger colonies of *H. glaber* as compared to other highly social mole-rats like *F. damarensis* (75 vs. 12 individuals mean value, respectively, Table 1), may be relevant for the reduction of physical effort per capita during digging, and therefore lowering the need for structural specialization. It is known that increased group size lowers the costs of foraging in the social *C. hottentotus*[71], and that the increased number of non-breeding subordinates in *F. damarensis* is associated with reductions in the workload of "queens"[72], thus suggesting that the number of working (e.g., digging) individuals in the colony may influence morphology. A similar "energetic" analogy for group size was utilized by Berkovitz & Faulkes[73] to interpret the similar incisor growth rates found among naked mole-rats, laboratory rodents, and lagomorphs. The latter authors suggested that *H. glaber* do not show particularly increased rates of incisor growth to compensate for their chisel-tooth digging behavior because their social cooperative behavior allows them to distribute digging activities among a large workforce, thus preventing extreme wearing of their incisors[73]. It is important to mention that the independent evolution of highly cooperative systems in *Heterocephalus* and *Fukomys*, with distinct ancestral components hinders direct functional and phenotypic correspondence among these social taxa, so that it is reasonable to expect their phenotypes and developmental pathways to differ.

In conclusion, our data suggests that the development of appendicular adaptations in mole-rats may have resulted from increased demands to burrowing, although forelimbs and hind limbs evolved independently, probably associated with different levels of fossorial specialization. Several factors associated with the paedomorphic development and hyperspecialized chisel-tooth digging of *H. glaber*, as well as with the formation of organized digging sequences, large colony sizes and peculiar soil disposal, may have had an important role on the comparatively reduced fossorial specialization of their limb bones.

## Methods

According to IUCN Red List[74], 22 species (and 6 genera) of African mole-rats (Bathyergidae) are currently recognized. We investigated six genera and eight species (Table 1 and Supplementary Table 1). A total of 382 specimens including both sexes were classified in multiple ontogenetic stages, mostly comprising adults (Supplementary Methods). The majority of specimens were wild-caught, although some individuals of *F. damarensis* and *F. mechowii*, and all individuals of *H. glaber* come from captive colonies. Stylopods (femur and humerus) ($n = 703$) and zeugopods (ulna, tibia and fibula) ($n = 1000$) were analyzed. Most bones were dissected and skeletonized, while entire perinatal individuals were cleared and stained with Alcian blue (cartilage) following a pH adjustment of 2.5 and Alizarin red (bone) as described by Ovchinnikov[75]. Additionally, complete appendicular skeletons of individuals of multiple ages of *H. africaeaustralis* (Hystricidae, $n = 18$) and one adult of *P. typicus* (Petromuridae) from the Iziko SA Museum (Cape Town, South Africa) were used for anatomical comparisons (Fig. 6c). The adult skeletal phenotype of *T. swinderianus* was obtained from previous anatomical descriptions and illustrations reported by Onwuama et al.[65,76], which are based on 12 individuals. From the larger sample of bathyergids, 294 bones pertaining to 75 individuals from all species, except *Bathyergus janetta*, were scanned for determination of complex bone superstructures (e.g., fusion of tibia-fibula), particularly those of small specimens. Scans were performed at the micro-focus X-ray tomography facility (MIXRAD) of the South African Nuclear Energy Corporation (NECSA), Pelindaba, at resolutions ranging from 23 μm (small specimens) to 44 μm (larger specimens) isotropic voxel size using a Nikon XTH 225ST equipment[77]. 3D models were digitally rendered with Avizo v.9.0 software (Visualization Sciences Group Inc.).

**Bone superstructures and morpho-functional indices.** We qualitatively assessed the adult phenotype and early postnatal development of four (discrete) limb bone superstructures associated with fossoriality, which are straightforward to interpret and compare: the deltoid tuberosity (DT), olecranon process (OP), third trochanter (TT), and distal tibio-fibular fusion (DFTFi) (Supplementary Methods). Three morpho-functional indices that account for the level of functional specialization of the DT, OP and DFTFi were studied, the RDT, IFA, and TJI, respectively (Supplementary Methods; Supplementary Table 7). Additionally, to visualize the evolutionary and functional relevance of the DT among subterranean mammals, we carried out a comprehensive compilation of the humeral phenotype of 11 other fossorial rodent genera (25 species), as well as of 5 hystricid genera (5 species) as outgroups (Fig. 6a and Supplementary Table 6). Humeral morphology was included in a consensus tree for the phylogeny of 40 species of rodents (including 7 outgroup species for Bathyergidae) estimated with TreeAnnotator BEAST[78], using a subset of 100 birth-death node-dated completed trees obtained from www.vertlife.org[79].

**Phylogenetic varying effects regression.** Bayesian multilevel models were used to assess the scaling effect of three morpho-functional indices (RDT, IFA and TJI) with body mass (BM). Indices and BM were log-transformed and modeled as normally distributed random variables. A total of 247 individuals were analyzed (for IFA), but because *H. glaber* lacks a DT and DFTFi, this species was excluded from modeling RDT and TJI, resulting in a sample size of 151 individuals for the other two indices (Supplementary Table 7).

Instead of using species averages as input data, which is prone to bias due to measurement error[80], we used the data at their original individual level and let both intercept and slope to vary by species. In this way, we utilized the full sample and took advantage of partial pooling property of multilevel models, which adaptively regularizes not only estimates of each species' intercept and slope by informing parameter values simultaneously across all the species, but also estimates of parameters of their respective sampling distributions[80,81].

Moreover, unlike simple linear regression, multilevel models allow to model variability both within and between species[81]. To account for the phylogenetic effect, we used a consensus tree obtained from www.vertlife.org[79], including all seven bathyergid species in our sample. Details on model definition are presented in Supplementary Methods.

We then re-fitted all three models but with BM excluded, all else being equal. Fit of the models with and without BM for each index was then compared using Widely Applicable Information Criterion (WAIC)[82] to assess the relative importance of BM for the out-of-sample prediction. Posterior distributions for each parameter were obtained by running four chains of Hamiltonian Monte Carlo algorithm implemented in the statistical programming language Stan[83]. Each chain was run for 5000 iterations and with a 50% warm-up. Convergence was assessed by

Gelman-Rubin diagnostic and by the number of effective samples. All models were fitted with the rethinking package[80].

**Ancestral state reconstructions.** Ancestral states for three bone superstructures (DT, TT, and DFTFi) in Phiomorpha were reconstructed using a stochastic character mapping[84,85], implemented in the R package *phytools*[86]. Because OP is present in all taxa analyzed, it was excluded from the analysis. A character's history across the tree is sampled from a posterior distribution of possible character histories conditional on observed values at the tips, topology, and branch lengths. After the values at internal nodes have been assigned, a character's change along a given branch is then simulated, conditional on parameters of the transition rate matrix Q. We simulated 1000 maps averaging over the posterior sample of trees for Bathyergidae with the addition of three closely related species, *H. africaeaustralis*, *P. typicus*, and *T. swinderianus*. Simulated character state values at internal nodes are interpreted as posterior probability (PP). All statistical analyses were carried out in RStudio[87].

**Reporting summary.** Further information on research design is available in the Nature Research Reporting Summary linked to this article.

## Data availability
All the data generated and analysed in this study are included in this published article and its supplementary information files.

## Code availability
The code for all analyses is available at: https://github.com/gabrielsaffa/african_mole_rats.

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

## Acknowledgements

We are very grateful to Marcelo Sánchez-Villagra (Universität Zürich) for his support during the completion of this study. We thank Tim Clutton-Brock (Kalahari Meerkat Project, Kuruman Station) and Sabine Begall (Universität Duisburg-Essen, Germany) for providing mole-rat specimens. Dr. Ashwin Isaacs (Histology Laboratory, Division of Cell Biology, UCT) is thanked for technical assistance with the diaphonization procedures. We also thank Frikkie de Beer, Jakobus Hoffman, and Lunga Bam (NECSA-Pelindaba) for their kind help and assistance during the microtomographic acquisitions. Denise Hamerton (Curator), Jofred Opperman (Collections Manager) and Gabriel Lukoji (Assistant Collections Manager) are also acknowledged for kindly giving access to pet-romurid and hystricid specimens at the Iziko SA Museum (Cape Town, South Africa), as well as to Jan Robovsky for access to hystricid comparative material at the Department of Zoology, University of South Bohemia. Hynek Burda, Helder Gomes Rodrigues, Daniel Galiano, Ondřej Mikula, Camilo López-Aguirre, Alberto Valenciano, and Laura Bento Da Costa are also thanked for their useful comments on the manuscript and discussions that helped considerably to improve the quality of this study. This project was supported by Becas Chile, Government of Chile (CONICYT, 72160463), National Research Foundation (NRF-117716), the SARChI chair of Mammalian Behavioural Ecology and Physiology (DST-NRF-64756), and the Czech Science Foundation project GAČR (20-10222S). Financial support for the research trip to Kenya of J.U.M.J. was provided by The National Geographic Society (Grant 2111), whereas funding support for the maintenance of the original NMR colonies was provided by the University of Cape Town and the South African National Research Foundation (NRF). DST-NRF is acknowledged for the financial support (Grant # UID23456) to establish the MIXRAD micro-focus X-ray tomography facility at Necsa.

## Author contributions

G.M.S., N.C.B., R.Š., and A.C. designed the study; N.C.B., R.Š., and J.U.M.J. provided mole-rat specimens; G.M.S., J.U.M.J., and N.C.B. conducted experiments; G.M.S. and G.S. analyzed the data and created figures; G.M.S. acquired images and prepared the first draft of the manuscript; all authors read, edited, and approved the final manuscript.

## Competing interests

The authors declare no competing interests.
