## [Peer Review File · Communications Biology]

Reviewers' comments:

Reviewer #1 (Remarks to the Author):

This is a very interesting investigation of appendicular anatomy across the Bathyergidae. It is always nice to see postcranial morphology studies in rodents as they are quite rare and overshadowed by the much more expansive literature on the skull. I was also very impressed with the sample amassed by the authors, both in terms of sheer numbers and the range of life stages represented. It was interesting, if perhaps not surprising, to find out that the naked mole-rat is the odd one out compared to other bathyergids, as it is in so many other respects.

My feeling is that this is a solid piece of work with some great figures. I do not have any major concerns. I have provided comments on an annotated PDF, but they are all relatively minor.

Philip Cox
University of York

Reviewer #2 (Remarks to the Author):

The manuscript entitled "Fossorial adaptations in African mole-rats (Bathyergidae), the unique appendicular phenotype of naked mole-rats" presents a comprehensive anatomical account of the evolutionary and ontogenetic diversity in the postcranial skeleton in a clade of specialised burrowing mammals. The authors have compiled an impressive dataset, capturing within-species diversity across eight representative taxa of the family Bathyergidae. They focus much of the discussion on one finding; the sister taxon to the main clade, the naked mole rat, is quite distinct from the other species and may represent an ancestrally "less specialised" phenotype, or alternatively relaxed selection due to their distinct behaviour. They posit some hypotheses in the introduction relating to differences in the skeleton between taxa that have different digging modes. The data do not support these, and instead suggest all of the taxa (except the naked mole rat) have similar phenotypes. These results are discussed from an adaptation and selection perspective.

These are fascinating creatures, and the naked mole rat is a model organism, so this paper has the potential to have a broader appeal to evolutionary biologists. However, in its current format, the evolutionary discussion is diluted with a large amount of specialist anatomical descriptions that are better suited to a specialist morphology journal.

The naked mole rat does appear to be the main focus of the results and discussion, but is not introduced as a focus in the introduction (only briefly mentioned in the "spoiler", line 91). It is not clear as to why it is presented like this, and makes for a confusing read.

The study is mostly qualitative, describing the anatomical differences bone by bone across the taxa. The quantitative data are buried late in the results section, and also discussed late in the discussion, and mainly presented to refute the influence of body size on these variables. My main concern is that these variables are not used to their fullest. There is striking within-species variation of each index but this is never explained. These data could have been mapped to a phylogeny and examined with modern phylogenetic comparative methods with some statistical rigour (despite the low species numbers). But instead the authors discuss anatomical characters over the tree without a formal analysis.

The results section as it stands does not follow the structure posed by the introduction or discussion; they seem to be from different manuscripts. In particular, the results narrative bounces between different figures, making it hard to follow. I think there is something very interesting in this story, where the ontogenetic sequences provide some indication of the evolutionary history. But it is hard to find given the descriptive and anatomically-detailed narrative of the results.

Overall, I think this paper is trying to cover too much detail for a journal with such wide, general readership. To remove most and hide away in supplementary materials would however do the work injustice. But at the very least, the results need to be rewritten to properly address the aims set out in the introduction, and lead to the main patterns being discussed.

The morphometric indices are underutilised and I encourage the authors to bring these to the forefront of the study, because they show the most promise to accurately estimate the biomechanical differences in an evolutionary context.

Reviews should not discuss what has not been done; but I urge the authors to consider at some future time a thorough morphometric analysis of these bones, where they can really critically address the morphological changes during ontogeny and evolution.

Minor comments

To make this manuscript accessible to a wide audience, the authors must explain what bathyergid is in the abstract, and stick to using the common name for the naked mole rat. Throughout the manuscript, there is a back and forth between using the Latin names and the common names, which is confusing to follow.

The taxonomy of bathyergids is not my speciality, but a search of this family name suggests the naked mole rat has been moved into its own family. Please address the taxonomic scheme you are using here.

Line 58 – need to say how many species, and what kind of coverage this is of the total taxonomic diversity in the family. Generic diversity is rather meaningless.

Line 79 - Scratch-digging evolved only once, so you need to be cautious when making adaptive assertions in your study of their phenotype.

Figure 1 – DC not explained; only 7 taxa shown, but does not say why; using a-e in caption is confusing because this does not refer to any letters or parts of the figure; yellow arrows very subjective – how has this information been captured?

Figure 5 – A very nicely arranged figure. Asterisks not explained. This is a key result, but lost among so much of the text and other figures/results.

Some awkward grammatical issues throughout; suggest some proof-reading help is requested.

Reviewer #3 (Remarks to the Author):

Dear Authors

This manuscript has an interesting approach to the study of the anatomy of the limbs of Bathyergidae. The major aim of the work is to assess the morphological diversity and early development of limb fossorial adaptations of African mole-rats. It uses a large sample of species and specimens (so the anatomical traits were well-studied regarding their potential variability) and a phylogenetic framework for comparisons. Also, stained specimens of newborns were analyzed to understand the first steps in the ontogeny of morphological changes, giving interesting clues about this topic (This is clearly the more striking point of the manuscript).

I think the sample could have been used better, with more detailed descriptions and images (I am an anatomist so this is what I enjoy to see!). And there are some things that they have to explain to better substantiate some choices and to better explained the materials and methods used.

There could be a more in-depth discussion of some anatomical traits (to take advantage of the impressive sample!). Phylogenetic framework should be used to take into account the phylogenetic structure in the relation between indices and BM. This is a nice work to seen published, after some work on the items I mentioned.

I have several comments and some suggestions to text writing in the attached word docx file (I apologize by the names in comments!)

Response to Editors and Referees

Dear Editors and Referees,

Thank you very much for handling our manuscript and making valuable corrections and suggestions to our study. Your time and effort are very much appreciated. We have addressed all the corrections and provide a new updated document. Essentially, the major changes made in the manuscript correspond to the incorporation of phylogenetic and body mass corrections, as well as new Figures. If the editor and reviewers consider it appropriate, we have also provided a list of abbreviations at the beginning of the manuscript. Because all of these changes, the order and flow of sentences in our manuscript has changed considerably. Consequently, Results and Discussion sections have been modified accordingly and greatly improved. We have also included a new co-author (G. Šaffa) in the study, who helped with the realization of multiple statistical procedures requested by the referees. Below (*in blue*) are our detailed answers to the points raised by the reviewers.

Please do not hesitate to contact us if you need further information or clarification on any of the aspects raised by the reviewers.

Best regards,

Germán Montoya-Sanhueza (PhD)

(On behalf of all co-authors)

Postdoctoral Fellow
Department of Zoology
University of South Bohemia
(České Budějovice, Czechia)

Editor comments

- Reviewers 2 and 3 are keen to see some additional quantitative analysis, in particular with respect to character evolution across the phylogenetic tree.

RE: We added phylogenetic corrections and an assessment of the effects of body mass for the analysis of morpho-functional indices, as well as added the reconstruction of character states for bone superstructures. See the following sections in Results (and Material and Methods):

- Effects of Body Mass and Phylogeny on Morpho-functional Indices
- Ancestral State Reconstructions

- The reviewers would also like to see some revisions and clarifications made to the results and discussion section, particularly with respect to strengthening the narrative of the evolutionary discussion and with more emphasis given to the quantitative aspects of the study.

RE: Based on the new quantitative analyses, the Results and Discussion sections are greatly improved. The last section in Discussion (*Evolution of Fossorial Adaptations in Bathyergidae and naked mole-rats*) has been considerably modified to support our quantitative findings. We have also included relevant information on the fossil record of bathyergids to strength our Discussion.

- Please note that I am satisfied that the anatomical descriptions are suitable in their current form provided that the key evolutionary results and discussion are clear and form a coherent narrative.

RE: Thank you very much. Because we had contradictory requests by the referees (Referee 2 asked for less anatomical descriptions, whereas Referee 3 asked for more anatomical descriptions), it is important to point out that we have **four** other ongoing studies about the limb morphology of African mole-rats:

1. **Manuscript (Ms) accepted in Journal of Mammalian Evolution; Montoya-Sanhueza et al. (2022) Developmental Plasticity in the Ossification of the Proximal Femur of Heterocephalus glaber (Bathyergidae, Rodentia) <https://doi.org/10.1007/s10914-022-09602-y>**
2. **Ms in second round of revisions, which includes full assessment of 17 morpho-functional indices: Montoya-Sanhueza et al. Functional anatomy and disparity of the postcranial skeleton of African mole-rats (Bathyergidae); the special case of the naked mole-rat**
3. **Ms in preparation, about growth patterns in the family (Allometry).**
4. **Ms in preparation, on their fore- and hind limb muscle architecture.**

We are therefore aware of the importance of describing the musculoskeletal anatomy of this rodent family in detail and we would prefer to maintain the actual level of anatomical descriptions presented in the current study as is, although we have included some figures to improve the visualization of the described traits, e.g. Supplementary Figure 1.

Reviewer #1 (Remarks to the Author):

This is a very interesting investigation of appendicular anatomy across the Bathyergidae. It is always nice to see postcranial morphology studies in rodents as they are quite rare and overshadowed by the much more expansive literature on the skull. I was also very impressed with the sample amassed by the authors, both in terms of sheer numbers and the range of life stages represented. It was interesting, if perhaps not surprising, to find out that the naked mole-rat is the odd one out compared to other bathyergids, as it is in so many other respects.

My feeling is that this is a solid piece of work with some great figures. I do not have any major concerns. I have provided comments on an annotated PDF, but they are all relatively minor.

RE: Thank you very much for your kind words. We have addressed and corrected all the comments made by the reviewer in the provided document.

Reviewer #2 (Remarks to the Author):

The manuscript entitled “Fossorial adaptations in African mole-rats (Bathyergidae), the unique appendicular phenotype of naked mole-rats” presents a comprehensive anatomical account of the evolutionary and ontogenetic diversity in the postcranial skeleton in a clade of specialised burrowing mammals. The authors have compiled an impressive dataset, capturing within-species diversity across eight representative taxa of the family Bathyergidae. They focus much of the discussion on one finding; the sister taxon to the main clade, the naked mole rat, is quite distinct from the other species and may represent an ancestrally “less specialised” phenotype, or alternatively relaxed selection due to their distinct behaviour. They posit some hypotheses in the introduction relating to differences in the skeleton between taxa that have different digging modes. The data do not support these, and instead suggest all of the taxa (except the naked mole rat) have similar phenotypes. These results are discussed from an adaptation and selection perspective.

These are fascinating creatures, and the naked mole rat is a model organism, so this paper has the potential to have a broader appeal to evolutionary biologists. However, in its current format, the evolutionary discussion is diluted with a large amount of specialist anatomical descriptions that are better suited to a specialist morphology journal.

The naked mole rat does appear to be the main focus of the results and discussion, but is not introduced as a focus in the introduction (only briefly mentioned in the “spoiler”, line 91). It is not clear as to why it is presented like this, and makes for a confusing read.

The study is mostly qualitative, describing the anatomical differences bone by bone across the taxa. The quantitative data are buried late in the results section, and also discussed late in the discussion, and mainly presented to refute the influence of body size on these variables. My main concern is that these variables are not used to their fullest. There is striking within-species

variation of each index but this is never explained. These data could have been mapped to a phylogeny and examined with modern phylogenetic comparative methods with some statistical rigour (despite the low species numbers). But instead the authors discuss anatomical characters over the tree without a formal analysis.

Re: Thank you very much. We have used our morphological data accordingly to test the influence of body size and phylogeny in our results, as well as to reconstruct ancestral stages.

The results section as it stands does not follow the structure posed by the introduction or discussion; they seem to be from different manuscripts. In particular, the results narrative bounces between different figures, making it hard to follow. I think there is something very interesting in this story, where the ontogenetic sequences provide some indication of the evolutionary history. But it is hard to find given the descriptive and anatomically-detailed narrative of the results.

Overall, I think this paper is trying to cover too much detail for a journal with such wide, general readership. To remove most and hide away in supplementary materials would however do the work injustice. But at the very least, the results need to be rewritten to properly address the aims set out in the introduction, and lead to the main patterns being discussed.

Re: we have rewritten a major part of the Results and Discussion sections to include the requests of the reviewers.

The morphometric indices are underutilised and I encourage the authors to bring these to the forefront of the study, because they show the most promise to accurately estimate the biomechanical differences in an evolutionary context.

Re: we have improved our analysis and clarified the interpretations in the main text.

Reviews should not discuss what has not been done; but I urge the authors to consider at some future time a thorough morphometric analysis of these bones, where they can really critically address the morphological changes during ontogeny and evolution.

Re: Thank you very much, such morphometric analyses have already been done, and the MS is currently under corrections (see above).

Minor comments

To make this manuscript accessible to a wide audience, the authors must explain what bathyergid is in the abstract, and stick to using the common name for the naked mole rat. Throughout the manuscript, there is a back and forth between using the Latin names and the common names, which is confusing to follow.

Re: We have modified the abstract accordingly, and standardized the use of the Latin name *Heterocephalus glaber* in most of the text.

The taxonomy of bathyergids is not my speciality, but a search of this family name suggests the naked mole rat has been moved into its own family. Please address the taxonomic scheme you are using here.

Re: Thank you very much for the taxonomical observation. The authors of this study, which include renowned members of the mole-rat research community, agree on using the traditional classification of *Heterocephalus* as contained within the family Bathyergidae. The paper of Patterson & Upham (2014) positioning *Heterocephalus glaber* within Heterocephalidae is a taxonomical hypothesis, where the comparative “analysis” of the anatomy used by these latter authors is not exhaustive. For example, they attribute some skeletal features to be present in *H. glaber* and all other Bathyergoidea (p. 946 in Patterson & Upha, 2014), although such traits were never examined in specimens or compared with other species (this is one of the main results of our study).

In our experience, the issue of the supra-taxonomic position of this taxon among African mole-rats has been premature, lacking fundamental data from the grounds of morphology, anatomy and development, as well as from the fossil record. Further empirical testing would improve our understanding on this issue. Additional details about the arbitrary family-level classification of *H. glaber* within Heterocephalidae can be found in two recent publications from two of our co-authors, Braude et al. (2020) and Buffenstein et al. (2022).

We follow the more recent taxonomical approach of Bryja et al. (2018), Visser et al. (2019) and Uhrová et al. (2022) which significantly extended the sampling, analysed nuclear markers and cytochrome b sequences, re-analysed mtDNA, and included detailed geographic information for all six genera of this African subterranean rodent family.

References

Braude, S., Holtze, S., Begall, S., Brenmoehl, J., Burda, H., Dammann, P., Del Marmol, D., Gorshkova, E., Henning, Y. & Hoeflich, A. (2021). Surprisingly long survival of premature conclusions about naked mole-rat biology. *Biological Reviews of the Cambridge Philosophical Society* 96, 376–393

Bryja, J., Konvičková, H., Bryjová, A. et al. 2018. Differentiation underground: Range-wide multilocus genetic structure of the silvery mole-rat does not support current taxonomy based on mitochondrial sequences. *Mamm Biol* 93, 82–92.

Buffenstein R, Amoroso V, Andziak B, Avdieiev S, Azpurua J, Barker AJ, Bennett NC, Briño-Enríquez MA, Bronner GN, Coen C, Delaney MA, Dengler-Crish CM, Edrey YH, Faulkes CG, Frankel D, Friedlander G, Gibney PA, Gorbunova V, Hine C, Holmes MM, Jarvis JUM, Kawamura Y, Kutsukake N, Kenyon C, Khaled WT, Kikusui T, Kissil J, Lagestee S, Larson J, Lauer A, Lavrenchenko LA, Lee A, Levitt JB, Lewin GR, Lewis Hardell KN, Lin TD, Mason MJ, McCloskey D, McMahon M, Miura K, Mogi K, Narayan V, O'Connor TP, Okanoya K, O'Riain MJ, Park TJ, Place NJ, Podshivalova K, Pamenter ME, Pyott SJ, Reznick J, Ruby JG, Salmon AB, Santos-Sacchi J, Sarko DK, Seluanov A, Shepard A, Smith M, Storey KB, Tian X, Vice EN, Viltard M, Watarai A, Wywiał E, Yamakawa M, Zemlemerova ED, Zions M, Smith ESJ. 2021 The naked truth: a comprehensive clarification and classification of current 'myths' in naked mole-rat biology. *Biol Rev Camb Philos Soc*. doi: 10.1111/brv.12791.

Patterson BD, Upham NS. 2014. A newly recognized family from the Horn of Africa, the Heterocephalidae (Rodentia: Ctenohystrica). *Zoological Journal of the Linnean Society* 172: 942–963.

Uhrová M, Mikula O, Bennett NC, et al. 2022. Species limits and phylogeographic structure in two genera of solitary African mole-rats *Georychus* and *Heliophobius*. *Molecular Phylogenetics and Evolution*;167:107337.

Visser, J. H., Bennett, N. C. & van Vuuren, B. J. (2019). Phylogeny and biogeography of the African Bathyergidae: a review of patterns and processes. PeerJ 7, e7730

Line 58 – need to say how many species, and what kind of coverage this is of the total taxonomic diversity in the family. Generic diversity is rather meaningless.

Re: we have included in Material and Methods

Line 79 - Scratch-digging evolved only once, so you need to be cautious when making adaptive assertions in your study of their phenotype.

Re: corrected.

Figure 1 – DC not explained; only 7 taxa shown, but does not say why; using a-e in caption is confusing because this does not refer to any letters or parts of the figure; yellow arrows very subjective – how has this information been captured?

Re: most suggestions were corrected accordingly. We only showed one species per genus, and for the case of *Fukomys* we included the largest species of this lineage because its significant difference in body size with the rest of the bathyergids. We kept the yellow arrows indicating the direction of bone thickening in the humerus, since these are important to highlight the narrow diaphysis of *H. glaber*.

Figure 5 – A very nicely arranged figure. Asterisks not explained. This is a key result, but lost among so much of the text and other figures/results.

Re: all corrected.

Some awkward grammatical issues throughout; suggest some proof-reading help is requested.

Re: The manuscript was reviewed by all co-authors including NB, ACh and JJ whom are native English speakers.

Reviewer #3 (Remarks to the Author):

Dear Authors

This manuscript has an interesting approach to the study of the anatomy of the limbs of Bathyergidae. The major aim of the work is to assess the morphological diversity and early development of limb fossorial adaptations of African mole-rats. It uses a large sample of species and specimens (so the anatomical traits were well-studied regarding their potential variability) and a phylogenetic framework for comparisons. Also, stained specimens of newborns were analyzed to understand the first steps in the ontogeny of morphological changes, giving interesting clues about this topic (This is clearly the more striking point of the manuscript).

I think the sample could have been used better, with more detailed descriptions and images (I am an anatomist so this is what I enjoy to see!). And there are some things that they have to

explain to better substantiate some choices and to better explained the materials and methods used. There could be a more in-depth discussion of some anatomical traits (to take advantage of the impressive sample!). Phylogenetic framework should be used to take into account the phylogenetic structure in the relation between indices and BM. This is a nice work to seen published, after some work on the items I mentioned.

I have several comments and some suggestions to text writing in the attached word docx file (I apologize by the names in comments!)

Alicia Álvarez

Re: Thank you very much for your detailed review. As mentioned above, we have four ongoing investigations on the morphological aspects of the family Bathyergidae, so we expect that the more detailed anatomical description of this family will be available soon.

We have also addressed all the suggestions made in the document provided by the reviewer. Please let us know if you have any other request or doubt about our study.

REVIEWERS' COMMENTS:

Reviewer #2 (Remarks to the Author):

Thank you for your detailed and considered revisions. I am satisfied with how my earlier comments, and those of the other reviewers, have been addressed and believe this is an excellent manuscript that is an important advance for the field. I have no further comments to make and look forward to seeing this and those other publications on this system published.

Reviewer #3 (Remarks to the Author):

The manuscript "Fossorial adaptations in African mole-rats (Bathyergidae) and the unique appendicular phenotype of naked mole-rats" by Montoya-Sanhueza and colleagues encompasses a study of the anatomy of the limbs of members of this family. The major aim of the work is to assess the morphological diversity and early development of fossorial adaptations of limbs of African mole-rats. It includes all bathyergid genera and several species, represented by a large sample of specimens. It uses a phylogenetic framework for anatomical descriptions and comparisons, ancestral reconstruction of relevant postcranial traits and allometric effects on functional indices. Stained specimens of newborns were analyzed to understand the first steps in the ontogeny of morphological changes, giving interesting clues about this topic. It discusses the appendicular fossorial adaptations of bathyergid species in the context of the family and in a broader context, giving some examples from other rodent groups and the fossil record of Bathyergidae. Morphogenesis of fossorial adaptations is also discussed.

The manuscript "Fossorial adaptations in African mole-rats (Bathyergidae) and the unique appendicular phenotype of naked mole-rats" by Montoya-Sanhueza and colleagues is now a more solid work, with analytical background for several assertions that needed that. As for the original version of the manuscript, it is a very interesting work on the anatomy of rodents, with an impressive sample, and that focuses, in part, in ontogeny, a dimension of morphological diversity understudied.

I have some few minor comments:

In Table 1 (in "9493_1_art_file_341190_r8rljk" file) it is referenced the size of *B. suillus* as "783 gr" while in the text it is said "to >2 kg in the Cape dune mole-rat *Bathyergus suillus*". Please, check.

Figure 4. I think it is not clear what "Posterior predictions" means in the context of a legend of a figure. Maybe it could start with "scaling effect" (of body mass on functional indices)?

line 457. "Two" in the phrase "The ancestral reconstructions including the two non-subterranean closest living relatives of bathyergids, the Petromuridae, Thryonomyidae, and Hystricidae," should be replaced by "three"?

line 630. "Indices were log-transformed". BM was also log-transformed according figure 4, so it should be indicated in text too.

line 643. Why was used the VertLife phylogeny used here instead that of Uhrová et al. that is indicated in line 186 as that used for phylogenetic relationships?

Response to Editors and Referees

Dear Editors and Referees,

Thank you very much for handling our manuscript and making valuable corrections and suggestions to our study. Your time and effort are very much appreciated. We have addressed all the corrections of the second round, including the changes requested for the formatting of the manuscript (following **Final Revision Instructions**).

We provide a new updated document, as well as the corresponding track changes for this. For a better comprehension and understanding of our interpretations, we have also corrected several typos and grammar mistakes throughout the manuscript, as well as modified some paragraphs in the Results and Discussion section (see “Track Changes document”). We also reduced the size of ‘panel c’ in Figure 6, for better reading of the phylogenetic tree, and interchanged the reference #68 by a more appropriate reference for our statements on pup growth rates (i.e. we deleted “Bennett et al 1991” and added “O’Riain & Jarvis 1998”). See below (*in blue*) our detailed answers to the points raised by the reviewers.

Please do not hesitate to contact us if you need further information or clarification on the changes performed.

Best regards,

Germán Montoya-Sanhueza (PhD)

(On behalf of all co-authors)

Postdoctoral Fellow
Department of Zoology
University of South Bohemia
(České Budějovice, Czechia)

REVIEWERS' COMMENTS:

Reviewer #2 (Remarks to the Author):

Thank you for your detailed and considered revisions. I am satisfied with how my earlier comments, and those of the other reviewers, have been addressed and believe this is an excellent manuscript that is an important advance for the field. I have no further comments to make and look forward to seeing this and those other publications on this system published.

Thank you very much for your kind words and the new revision of our manuscript.

Reviewer #3 (Remarks to the Author):

The manuscript “Fossorial adaptations in African mole-rats (Bathyergidae) and the unique appendicular phenotype of naked mole-rats” by Montoya-Sanhueza and colleagues encompasses a study of the anatomy of the limbs of members of this family. The major aim of the work is to assess the morphological diversity and early development of fossorial adaptations of limbs of African mole-rats. It includes all bathyergid genera and several species, represented by a large sample of specimens. It uses a phylogenetic framework for anatomical descriptions and comparisons, ancestral reconstruction of relevant postcranial traits and allometric effects on functional indices. Stained specimens of newborns were analyzed to understand the first steps in the ontogeny of morphological changes, giving interesting clues about this topic. It discusses the appendicular fossorial adaptations of bathyergid species in the context of the family and in a broader context, giving some examples from other rodent groups and the fossil record of Bathyergidae. Morphogenesis of fossorial adaptations is also discussed.

The manuscript “Fossorial adaptations in African mole-rats (Bathyergidae) and the unique appendicular phenotype of naked mole-rats” by Montoya-Sanhueza and colleagues is now a more solid work, with analytical background for several assertions that needed that. As for the original version of the manuscript, it is a very interesting work on the anatomy of rodents, with an impressive sample, and that focuses, in part, in ontogeny, a dimension of morphological diversity understudied.

Thank you very much for your words and new revision. This is very much appreciated.

I have some few minor comments:

In Table 1 (in “9493_1_art_file_341190_r8rljk” file) it is referenced the size of *B. suillus* as “783 gr” while in the text it is said “to >2 kg in the Cape dune mole-rat *Bathyergus suillus*”. Please, check.

RE: We believe it is important to mention the maximum body size of this species in the main text to have an idea of the body size range within the family. The table only illustrates mean body sizes.

Figure 4. I think it is not clear what “Posterior predictions” means in the context of a legend of a figure. Maybe it could start with “scaling effect” (of body mass on functional indices)?

RE: we have modified the caption.

line 457. “Two” in the phrase “The ancestral reconstructions including the two non-subterranean closest living relatives of bathyergids, the Petromuridae, Thryonomyidae, and Hystricidae,” should be replaced by “three”?

RE: corrected.

line 630. “Indices were log-transformed”. BM was also log-transformed according figure 4, so it should be indicated in text too.

RE: corrected.

line 643. Why was used the VertLife phylogeny used here instead that of Uhrová et al. that is indicated in line 186 as that used for phylogenetic relationships?

RE: The phylogeny used for formal analyses is the one obtained from VertLife. The simplified phylogeny showed in line 186 is based on Uhrová et al., and is showed for schematic purposes only. Both phylogenies have the same topology though.

Best regards,
Alicia Álvarez